# MicroRNAs in Pulmonary Hypertension, from Pathogenesis to Diagnosis and Treatment

**DOI:** 10.3390/biom12040496

**Published:** 2022-03-24

**Authors:** Junhua Xu, John Linneman, Yanfeng Zhong, Haoyang Yin, Qinyi Xia, Kang Kang, Deming Gou

**Affiliations:** 1Shenzhen Key Laboratory of Microbial Genetic Engineering, Vascular Disease Research Center, Department of Biotechnology, College of Life Sciences and Oceanography, Shenzhen University, Shenzhen 518060, China; bzxjhbio@gmail.com (J.X.); yanfengzhong85@gmail.com (Y.Z.); yinhaoyang0224@gmail.com (H.Y.); xiaqy20@gmail.com (Q.X.); kangkang@szu.edu.cn (K.K.); 2Guangdong Provincial Key Laboratory of Regional Immunity and Disease, College of Life Sciences and Oceanography, Shenzhen University, Shenzhen 518060, China; 3Key Laboratory of Optoelectronic Devices and Systems of Ministry of Education and Guangdong Province, College of Optoelectronic Engineering, Shenzhen University, Shenzhen 518060, China; 4School of Medicine, Washington University, 12275, St. Louis, MO 63110, USA; jmlinneman@wustl.edu

**Keywords:** pulmonary hypertension, microRNAs, diagnostic biomarker, microRNA target drugs, off-target effects

## Abstract

Pulmonary hypertension (PH) is a fatal and untreatable disease, ultimately leading to right heart failure and eventually death. microRNAs are small, non-coding endogenous RNA molecules that can regulate gene expression and influence various biological processes. Changes in microRNA expression levels contribute to various cardiovascular disorders, and microRNAs have been shown to play a critical role in PH pathogenesis. In recent years, numerous studies have explored the role of microRNAs in PH, focusing on the expression profiles of microRNAs and their signaling pathways in pulmonary artery smooth muscle cells (PASMCs) or pulmonary artery endothelial cells (PAECs), PH models, and PH patients. Moreover, certain microRNAs, such as miR-150 and miR-26a, have been identified as good candidates of diagnosis biomarkers for PH. However, there are still several challenges for microRNAs as biomarkers, including difficulty in normalization, specificity in PH, and a lack of longitudinal and big sample-sized studies. Furthermore, microRNA target drugs are potential therapeutic agents for PH treatment, which have been demonstrated in PH models and in humans. Nonetheless, synthetic microRNA mimics or antagonists are susceptible to several common defects, such as low drug efficacy, inefficient drug delivery, potential toxicity and especially, off-target effects. Therefore, finding clinically safe and effective microRNA drugs remains a great challenge, and further breakthrough is urgently needed.

## 1. Introduction

Pulmonary hypertension (PH) is a fatal and untreatable disease, ultimately leading to right heart failure and eventually death [1,2]. It is reported that various factors contribute to the pathogenesis of PH, such as genetic, epigenetic, and environmental factors. Of them, microRNAs (miRNAs) have been shown to play a critical role in PH pathogenesis. miRNAs are small, non-coding endogenous RNA molecules, consisting of ~21 to 25 nt [3,4]. A single miRNA can regulate hundreds of genes or proteins, and multiple miRNAs can regulate one protein [4].

miRNAs can influence various biological processes, such as cell survival, proliferation, and differentiation [5]. Briefly, as shown in Figure 1, miRNAs are transcribed and processed into several primary miRNAs (pri-miRNA), then transformed into pre-miRNA, and subsequently exit the nucleus and are processed into mature miRNA by Dicer. Finally, it is integrated into the RNA-induced silencing complex (RISC) where it binds to the 3′-untranslated region of mRNA and leads to mRNA degradation or inhibition of translation [6], resulting in reduced expression of a targeted gene. It has been estimated that about 1000 miRNAs regulate more than 30% of human genes [7]. Moreover, miRNAs have already been shown to be implicated in cancer, cardiovascular disease, and PH [6,8,9].

Herein, we have summarized the current understanding of miRNAs in the pathogenesis of PH and regulation of miRNAs by lncRNAs (long noncoding RNAs) in PH. Subsequently, we have discussed miRNAs as good candidates of diagnostic biomarkers for PH. Lastly, we have reviewed studies on therapeutic strategies of miRNAs for the treatment of PH.

## 2. miRNAs Correlated with PH

As is known, miRNAs play an important role in regulating fundamental biological processes, such as cell proliferation, differentiation, and apoptosis [10]. Additionally, changes in miRNA expression level can lead to various cardiovascular disorders, including atherosclerosis, peripheral vascular disease, and PH [11]. In recent years, in order to explore the role of miRNAs on PH, there have been numerous studies (Table 1) analyzing the expression profiles of miRNAs and their signaling pathway (Figure 2) in PASMCs or PAECs, PH models, and PH patients. Caruso et al. showed that the expression of Dicer was reduced in rat models induced by hypoxia, possibly indicating an overall down-regulation of miRNAs in PH [12]. Subsequently, miR-204 expression was found to be reduced in hypoxia- and monocrotaline (MCT)-induced PH rats [13]. Other miRNAs were found to be consistently up-regulated, such as miR-322 and miR-451, or down-regulated, such as miR-22, miR-30, and let-7f in hypoxia and MCT-induced PH [12]. Furthermore, our group has found that miR-223, miR-328, miR-339, miR-4632, miR-1181, and miR-1281 were down-regulated and prevented the development of PH. In contrast, miR-20a and miR-125a-5p were up-regulated and promoted the development of PH. Recently, it is reported that microRNA-146-5p promoted HPAEC cells proliferation under hypoxic conditions through targeting USP3, indicating that the microRNA-146-5p/USP3 axis was possibly served as a target for PH treatment [14]. Levels of miR-483-3p/-5p were down-regulated in the serum, lung ECs, and CD144-enriched EVs from iPAH patients and PH rats. Additionally, miR-483 might reduce experimental PH by inhibition of multiple adverse responses [15].

## 3. miRNAs That Promote Hypoxia- or MCT-Induced PH

### 3.1. The miRNA-143/145 Cluster

miR-143/145 (miRNA-143 and miR-145) are regulated by a common promoter in a polycistronic cluster and transcribed as one primary miRNA [16]. Initially, miR-143/145 is labeled as a SMC-specific miRNA, since it is most abundantly expressed in heart, vascular, and visceral SMCs [17,18] and has been shown to be necessary for maintenance of the contractile phenotype of SMCs [16,19]. Subsequently, it was reported that miR-143/145 expression is up-regulated in mice induced by hypoxia and in PH patients [13,20]. Moreover, PH Patients who carried bone morphogenetic protein receptor type II (BMPR2) mutations have increased miR-145 levels. However, the reduced expression of BMPR2 can also be induced miR-145 in human and mouse PASMCs, indicating that miR-145 is downstream of BMP signaling [20].

### 3.2. miRNA-21

miR-21(miRNA-21) is up-regulated in pulmonary arteries from several PH models and PH patients. Human PASMCs (HPASMCs) induced by hypoxia had a 3-fold increase in miR-21 expression, whereas the expression of its target genes, including programmed cell death protein 4 (PDCD4), Sprouty 2 (SPRY2), and peroxisome proliferator-activated receptor-α (PPARα), decreased. Moreover, overexpression of miR-21 in PASMCs caused cell proliferation and migration, leading to the development of PH [21]. Additionally, when hypoxia-induced PASMCS were treated with anti-miR-21, cell proliferation and migration were blocked [18], suggesting that the inhibition of miR-21 prevents and reverses hypoxia-induced PH [22].

In contrast, miR-21 plays a protective role from PH in human PAECs. It is reported that miR-21 expression is up-regulated by BMPR2 signaling when induced by hypoxia. miR-21 directly targets RhoB and suppresses its expression and kinase activity, leading to angiogenesis and vasodilatation inhibition. Moreover, overexpression of miR-21 decreases PDCD4 expression and protects mice from PH in the hypoxia/SU5416 model [23]. In addition, blocked expression of miR-21 promotes RhoB expression and its kinase activity, exacerbating hypoxia/SU5416-induced PH [24]. It is reported that knocking out miR-21 in mice causes activation of the PDCD4/caspase-3 axis in PAECs and leads to progressive PH. Therefore, miR-21 appears to act as a brake to inhibit the progression of PH in human PAECs.

To sum up, the apparently contradictory roles of miR-21 in PASMCs and PAECs above are possibly two results from opposite tests obtained in PH models after miR-21 knockdown [25].

### 3.3. miRNA-210

miR-210(miRNA-210) is induced in HPASMCs and mouse lungs via the HIF-1 pathway [26]. Our group has shown that miR-210 is the predominant miRNA induced by hypoxia in HPASMCs, and up-regulation of miR-210 suppresses E2F3 and inhibits apoptosis, resulting in hyperplasia of PASMCs [26,27].

In human PAECs, miR-210 was induced by hypoxia. It was reported that that miR-210 directly inhibited the expression of iron–sulfur cluster assembly proteins (ISCU1/2), controlling mitochondrial metabolic functions [28]. As was reported, mitochondrial dysfunction and iron homeostasis are key factors in PH development [29,30,31]. White et al. demonstrated that alterations in the miR-210-ISCU1/2 axis cause iron–sulfur deficiencies in vivo and promote PH [32].

### 3.4. miRNA-138

miR-138 (miRNA-138) expression is increased in PASMCs and PAECs induced by hypoxia [33,34]. It represses hypoxia-induced apoptosis in PASMCs by down-regulating Mst1, an Akt inhibitor which represses the PI3K/Akt pathway [33]. miR-138 is also essential in endothelial cell dysfunction, another hallmark of pulmonary arterial hypertension [34].

### 3.5. miRNA-17/92

The cluster miR-17/92 (miRNA-17/92) includes several miRNAs (miR-17-5p, miR-19a, miR-19b, miR-20a, and miR-92a). Expression of miR-17/92 was found to be up-regulated in PH induced by MCT or hypoxia [12]. Moreover, miR-17/92 is regulated by IL-6/signal transducer and activator of transcription 3 (STAT3) signaling, and its overexpression blocks BMPR2 signaling, a deficient pathway involved in PH [35]. Furthermore, miR-17/92 has been identified as an enhancer of hypoxic PH by targeting PDZ and LIM domain 5 (PDLIM5) in vivo. Additionally, miR-17/92 overexpression inhibited PDLIM5, consequently enhancing transforming growth factor β (TGF-β)/Smad2/3 activity in PASMCs in vitro [36].

### 3.6. miRNA-20a

Our group found that miR-20a (miRNA-20a) expression is up-regulated in both mice and HPASMCs induced by hypoxia, and miR-20a represses the protein kinase, cGMP-dependent, type I (PRKG1) gene via directly binding to the coding region of PRKG1. Moreover, miR-20a promotes the proliferation and migration of HPASMCs and prevents cell differentiation [47].

### 3.7. miRNA-125a-5p

miR-125a-5p (miRNA-125a-5p) was identified by our group and its expression was up-regulated by TGF-β1 or IL-6, which inhibited the proliferation and promoted apoptosis in PASMCs by targeting the TGF-β1 and IL-6/STAT3 signaling pathways. Moreover, miR-125a-5p-induced expression in vivo ameliorated the progression of MCT-induced PH [48].

## 4. miRNAs That Prevent PH

### 4.1. miRNA-124

The nuclear factor of activated T cells (NFAT) signaling pathway is related to cell proliferation and PH. Our group (Kang et al.) identified miR-124 (miRNA-124), which strongly inhibited NFAT activity, dephosphorylation, nuclear translocation, and NFAT-dependent transcription of IL-2. We have shown that miR-124 modulates the NFAT pathway by directly targeting nuclear factor of activated T cell cytoplasmic 1 (NFATc1), calmodulin-binding transcription activator 1 (CAMTA1), and polypyrimidine tract-binding protein 1 (PTBP1). Moreover, miR-124 is down-regulated in HPASMCs and mice lungs induced by hypoxia, and overexpression of miR-124 inhibited HPASMC proliferation [37]. Another report demonstrated that miR-124 expression levels were reduced in fibroblasts from experimental PH models and PH patients. Furthermore, reduced levels of miR-124 led to hyperproliferation and migration of fibroblasts, whereas overexpression of miR-124 blocked proliferation and migration of fibroblasts [38]. Therefore, the antiproliferative effects of miR-124 may be useful in developing treatments for PH.

### 4.2. miRNA-328

The levels of miR-328 (miRNA-328) were reduced in rats induced by hypoxia and pulmonary arteries (PAs) from PH patients [39]. miR-328 targeted insulin growth factor 1 receptor (IGF1R) and the L-type calcium channel α1C (CaV1.2). In rat PASMCs, overexpression of miR-328 induced cell apoptosis [39]. Moreover, transgenic mice overexpressing miR-328 in smooth muscle exhibited reduced right ventricular systolic pressure and wall thickness compared to wild-type mice in both normoxic and hypoxic conditions [39].

Furthermore, our group found that miR-328 was down-regulated via DNA methylation mediated by DNA methylation transferase 1 (DNMT1) in PASMCs stimulated with PDGF and in serum of PH patients associated with congenital heart disease (CHD-PAH). Mechanistically, miR-328 could inhibit PASMCs proliferation and migration by directly targeting the Ser/Thr-protein kinase-1 (PIM-1). This suggests that serum miR-328 is likely to serve as potential circulating biomarker for CHD-PAH diagnosis [40].

### 4.3. miRNA-424/503

It was reported that the expression of Apelin (APLN) was reduced in PAECs from PH patients, which caused down-regulation of miR-424/503 (miRNA-424/503) and consequently, promoted the expression of FGF2 and FGFR1 in PAECs [41,42]. Moreover, miR-424/503 was down-regulated in PAECs from PH patients, and consequently, FGF2/FGFR was up-regulated. In contrast, it was reported that miR-424/503 could inhibit proliferation of PAECs and PASMCs in PAEC-conditioned medium. Sustained elevation of miR-424/503 expression in the experimental PH models by intranasal lentiviral delivery prevented elevation of RVSP (right ventricular systolic pressure) and reversed vascular remodeling. These observations indicate that the restoration of miR-424/503 function may provide clinical benefit in PH [41].

### 4.4. miRNA-204

As was reported, PDGF, endothelin-1, and angiotension II could reduce miR-204 (miRNA-204) expression, thereby promoting proliferation and increased resistance to apoptosis through activating the Src-STAT3-NFAT pathway via up-regulation of SHP2 (Src activators) in PASMCs. Moreover, the miR-204 expression level was reduced in experimental PH rat models induced by hypoxia or MCT and in human PH patients. Conversely, miR-204 mimics can reverse MCT-induced PH [13]. It was reported that mesenchymal stromal cell-derived exosomes increased lung levels of miR-204, and consequently inhibited vascular remodeling and hypoxic pulmonary hypertension by suppressing STAT3 signaling [43].

### 4.5. miRNA-98

miR-98 (miRNA-98) level is down-regulated in hypoxia-induced PAECs, lung from mice induced by Sugen5416/hypoxia, and PAECs from PH patients. It was reported that miR-98 was positively regulated by PPARγ and directly targeted ET-1, which contributes to PAEC proliferation [44].

### 4.6. miRNA-193

It was reported that hydroxyeicozatetraenoic (HETE) and hydroxyoctadecadiene (HODE) acids induce the expression of the retinoid X receptor alpha which silences miR-193 (miRNA-193) expression by binding to its promoter. Moreover, miR-193 could directly target IGF1R, and miR-193 overexpression inhibited proliferation in PASMCs from PH. More importantly, intratracheal administration of miR-193 mimics could alleviate symptoms of PH in both MCT and hypoxia models [45].

### 4.7. miRNA-223

miR-223 (miRNA-223) was identified by our group and was significantly down-regulated in PASMCs and lungs of mice and rats induced by hypoxia. miR-223 could modulate cell proliferation, migration, and actomyosin reorganization in PASMCs through its novel targets, RhoB and MLC2, leading to vascular remodeling and the development of PH. It is also suggested that miR-223 could serve as a potential circulating biomarker and a small molecule drug for diagnosis and treatment of PH [49].

### 4.8. miRNA-339

Our group found that miR-339 (miRNA-339) was highly expressed in the cardiovascular system and was down-regulated by FGF2 and PDGF-BB in PASMCs, which was dependent on ERK and PI3K signaling. miR-339 was also down-regulated in the pulmonary arteries of rats with MCT-induced PH. Moreover, miR-339 can inhibit proliferation of PASMC by directly targeting fibroblast growth factor receptor substrate 2 (FRS2) [50].

### 4.9. miRNA-4632

Our group identified miR-4632 (miRNA-4632) and found that its expression was significantly decreased in HPASMCs induced by PDGF-BB and serum of patients with PH, which is mediated by histone deacetylation through the activation of PDGFR/PI3K/HDAC4 signaling. Moreover, miR-4632 inhibited the proliferation and promoted apoptosis in HPASMCs by directly targeting the c-JUN [51]. This suggests that serum miR-4632 is likely to serve as a potential circulating biomarker for PH diagnosis.

### 4.10. miRNA-1181

miR-1181 (miRNA-4632) was identified by our group, and its expression was found to be reduced in PASMCs induced with PDGF-BB and in the serum of adult and newborn patients with PH, which was mediated by PDGFR/PKC signaling. Moreover, miR-1181 prominently inhibited PASMCs proliferation and migration, and miR-1181 regulated the PASMCs proliferation by targeting STAT3, implying miR-1181 could serve as a therapeutic and diagnostic candidate for PH [52].

### 4.11. miRNA-1281

Our group found that miR-1281 (miRNA-1281) was down-regulated in human and rat PASMCs induced by PDGF-BB or hypoxia, which was mediated by the PI3K/DNMT1/miR-1281/HDAC4 axis. Moreover, miR-1281 expression was reduced in pulmonary arteries of rats induced by MCT and in the serum of patients with coronary heart disease and pulmonary arterial hypertension. Overexpression of miR-1281 suppressed cell proliferation and migration in PASMCs by directly targeting HDAC4. This suggests that miR-1281 may be used in diagnosis and therapy [53].

## 5. Regulation of miRNAs by lncRNAs in PH

Long none-coding RNAs (lncRNAs) are >200 bp in length and generally have no known protein-coding function [54]. In the past ten years, lncRNAs have been confirmed as critical regulators in normal body functions and the development of many diseases [54]. LncRNA expressions are influenced by a numbers of factor, such as hormones, nutrients, age, and sex. Therefore, lncRNA expression patterns are dependent on development stages, and are aberrantly regulated in a variety of diseases, including PH [54,55]. A total of 362 lncRNAs were identified as significantly differentially expressed genes by comparative microarray analysis between lncRNAs and mRNAs in lung tissues from a hypoxia-induced PH [56]. Among them, the metastasis-associated lung adenocarcinoma transcript 1 (MALAT1) significantly elevated under hypoxia. Knockdown of MALAT1 prevented the proliferation of ECs and inhibited vessel growth [57]. The expression of lncRNA MANTIS reduced in patients with idiopathic pulmonary arterial hypertension (iPAH) and in rats induced by MCT. Silencing of MANTIS inhibited angiogenic sprouting and alignment of endothelial cells in response to shear stress [58]. We have demonstrated that LnRPT (lncRNA regulated by platelet-derived growth factor and TGF-β) was down-regulated in pulmonary arteries from rats induced by MCT, and depletion of LnRPT promoted PASMCs proliferation [59].

Moreover, lncRNAs can also regulate the abundance or activity of other RNAs as competing endogenous RNAs (ceRNAs) [60,61]. For instance, H19 can elevate the expression level of Angiotensin II receptor Type 1 (AT1R) by sponging let-7b during the stimulation of PDGF-BB, subsequently leading to PH development by promoting PASMCs proliferation. Silencing H19 expression blocks pulmonary artery remodeling in MCT-induced mice and rat models of PH [62]. Similarly, by sponging and sequestering miR-124-3p.1, MALAT1 can up-regulate the transcription factor Krupple-like factor 5 (KLF5) and its downstream signaling, causing hyperactive cell cycle progression, and promotion of proliferation and migration of PASMCs [63].

## 6. Circulating miRNAs as Diagnostic Biomarkers for PH

In the early development of PH, there are no specific signs of illness. The symptom most frequently experienced is shortness of breath when taking exercises, which is often ignored until more severe symptoms occur, such as chest pain or syncope. Consequently, PH symptoms worsens until diagnosis, with over 2.8 years of delay [64]. At diagnosis, nearly 75% of PH patients have moderate to severe symptoms [65], showing severe pulmonary vascular remodeling and right ventricular (RV) dysfunction. To date, right heart catheterization remains the gold standard in PH diagnosis. However, it is invasive and a nightmare for PH patients. Therefore, it is of vital importance to develop less invasive diagnostic methods, namely biomarkers.

The general definition of a good diagnostic biomarker is that it should be easily measured, while it still reflects the pathophysiological process [66]. Up to now, there is no specific diagnosis biomarker for PH in the early stages. However, several circulating biomarkers combined with imaging tools are instructive for the diagnosis and prognosis of PH. Unfortunately, currently used biomarkers are not specific, for they are expressed only in severe stages of the disease, or possibly influenced by other metabolic functions [67].

Excitingly, it has been reported that miRNAs, which circulate freely in the mammalian blood, could serve as diagnostic markers for early diagnosis of acute myocardial infarction and heart failure in humans [68,69,70]. miRNAs are very stable in the blood and serum, but the molecular mechanisms remain elusive. Firstly, miRNAs are secreted as microvesicles, exosomes, or apoptotic bodies and are released into the blood [70,71,72,73]. Additionally, almost 90% of circulating miRNAs exist in a stable complex, primarily with the Ago2 protein [74], and the levels of circulating miRNAs measured are repeatable [75]. Up to now, it is reported that many miRNAs play an important role in PH, and they are tissue specific and regulated in time. Moreover, miRNAs can be detected in blood samples (whole blood or plasma) and analyzed by RT-qPCR [76]. This provides an easy and inexpensive diagnostic strategy [77]. Therefore, miRNAs are good candidates for biomarkers of PH [11].

It is reported that circulating miR-150 is reduced in the blood of PH patients, suggesting poor survival [78]. miR-150 was also significantly down-regulated in the lungs of MCT-induced rats. These studies indicate that miR-150 is an important biomarker for PH [78]. Moreover, Schlosser reported that circulating miR-26a is reduced in rats induced by MCT and in patients with iPAH, and that the levels of miR-26a are positively correlated with the 6 min walk distance, the rise in right ventricular systolic pressure, and right ventricular hypertrophy [79]. Furthermore, it is reported that a set of up-regulated plasma miRNAs, such as miR-21, miR-130a, miR-133b, miR-191, miR-204, and miR-208b, and a portion of down-regulated miRNAs, including miR-26a, miR-29c, miR-34b, and miR-451, can be used as candidates for diagnostic biomarkers of PH [77]. However, the change in miRNA expression in serum was not consistent with the lung tissue. For example, miR-451 was up-regulated in the lung tissue of the hypoxic PH (HPH) rat model, and miR-204 was down-regulated in the lung tissue of both the HPH rat model and iPAH patients [13].

At present, there are several challenges for using miRNAs as diagnostic biomarkers. These include the following: difficulty in normalization due to unavailability of reference molecules, specificity to PH, and a lack of longitudinal and big sample-sized studies [80,81]. Initially, frequently used methods of normalization for small RNAs were total RNA or 18S rRNA, but they are not applicable in plasma or serum miRNA analysis [82]. However, miRNA normalization is difficult for diagnosis, since some miRNAs are variable between individuals or from one study to another [76]. It is even worse considering some technical complications, including miRNAs extraction methods, contamination, and degradation [76]. Moreover, there is no specificity for miRNAs in diagnosis of PH. Most miRNAs down-regulated in PH are possibly down-regulated in other diseases. For instance, the expression levels of miR-124, miR-204, and miR-206 are also reduced in cancer [9,83,84], and miRNA cluster 17/92 and miR-210 expression levels rise in cancer [8,85,86] and cardiovascular disease [87,88,89]. Therefore, it is a great challenge for the clinical diagnosis of PH using miRNAs as diagnostic biomarkers. Furthermore, there is a lack of longitudinal or big sample-sized studies, and the results of miR-150 [78] or miR-26a [79] require additional verification using a larger population cohort. In addition, lncRNAs can also regulate the abundance or activity of miRNAs [90,91]; therefore, it adds to the difficulty in using miRNAs as diagnostic biomarkers.

## 7. miRNA Target Drugs as PH Therapeutic Agents

Considering the effects of miRNA target drugs in the prevention of experimental PH, one may speculate that miRNA target drugs are novel therapeutic agents for PH treatment in humans. An apparent benefit of employing miRNA is the ability of miRNAs to target multiple genes within a network, making them more efficient. In general, there are two main strategies adopted for therapeutic potential of miRNAs. One is substitution therapy which uses miRNA mimics when miRNAs are down-regulated. The other is inhibition therapy which applies anti-miRs in the case of up-regulated miRNAs. Subsequently, various strategies have been developed for miRNA-based therapeutics, such as antisense oligonucleotides, locked nucleic acid (LNA) antimiR, antagomirs and miR mimics, and miRNA expression vectors [92,93].

In the early experiment trial, the delivery of the miRNA target drugs remains a great challenge. It was once proposed that it is possible to achieve a wide distribution of miRNAs via intravenous, subcutaneous, or intraperitoneal deliveries. However, off-target effects of miRNA modulation frequently occurred, especially in the liver [94,95]. To minimize possible off-target effects, local delivery to lung vessels has been performed. Subsequently, it was found that intranasal delivery of lentivirus expressing miR-424 and miR-503 in rat models led to a decrease in RVSP and reduced expression of PCNA, FGF2, and FGFR1 [41]. Moreover, it was demonstrated that A-17, which blocks the expression of miR-17, improved heart and lung function in an experimental PH model by preventing lung vascular and right ventricular remodeling [96].

Subsequently, several therapy treatments have been reported in experimental PH models and human clinical trial. Intra-tracheal nebulization of miR-204 mimics combined with a transfection reagent decreased PA pressure, right ventricular hypertrophy (RVH), and PAs remodeling in the MCT rat model of PH [13]. In addition, the application of antagomiR-20a to activate BMPR2 signal could block cell proliferation in HPASMCs, and more importantly, inhibited vascular remodeling in experimental PH mice induced by hypoxia [97]. Moreover, the miR-122 inhibitor miravirsen (Santaris Pharma a/s, Copenhagen, Denmark) based on the LNA Drug Platform was the first miRNA-targeted drug to be used in a human clinical trial [93]. Its phase II study was carried out to evaluate the safety and efficacy of miravirsen in 36 patients with chronic hepatitis C virus (HCV) genotype 1 infection, which reduced HCV RNA levels durably in a dose-dependent manner (ClinicalTrials.gov number, NCT01200420) [98]. Moreover, we searched “microRNA and pulmonary hypertension” in ClinicalTrials.gov (search on 25 November 2021) and retrieved four clinical trials listed, three of which evaluated miRNA in PH, but only one (NCT00806312) was completed, which evaluated the miRNA profile and markers of inflammation in patients with PH.

Although miRNAs are an attractive new class of drugs possessing high therapeutic potential, there are still great challenges for therapy use of miRNAs. In general, synthetic miRNA mimics or antagonists are susceptible to common defects similar to RNAi drugs, including low drug efficacy, inefficient drug delivery, potential toxicity, and especially, off-target effects [99]. It has been shown that the antagomiRs effect is not immediate and antagomiRs work only after repeated application. It is possible that the abundance or activity of miRNAs are affected by lncRNAs [90,91]. Therefore, it is technologically challenging for the optimization and stabilization of the antagomiRs effects in the future [95]. Additionally, to promote drug delivery, local delivery to lung vessels is utilized, such as intranasal, nebulization, or intratracheal delivery. Moreover, there is no antidote to miRNA therapy which would allow for an immediate reversal of any potentially undesirable effects [95,97], and potential toxicity and various interactions of RNA-interference approaches with commonly used drugs [100] are currently unknown. Furthermore, off-target toxicity is most concerning. Off-target toxicity can be caused by both the innate and adaptive mammalian immune systems [101]. In addition, off-target effects might result from the overexpression of miRNA drugs, which possibly saturate the miRNA biogenesis machinery and change the expression of other nontargeted miRNAs [101,102].

To date, many approaches have been introduced to minimize the off-target effects of miRNAs. To avoid the drawbacks of directly targeting mature miRNAs, one method is to intervene in the processing of RNAs by targeting, for example, pri-mRNAs and pre-miRNAs [103]. Moreover, targeted delivery of miRNA drugs can limit unwanted effects [104]. It is reported that nanoparticle and antibody-based methods are fascinating cell- or tissue-specific delivery systems for miRNA drugs [101]. Furthermore, Hornung et al. found that immune activation is mediated by the nine nucleotides in the 3′ end of the sense strand of siRNA [105] and LNA modification of the 29th position of the sugar ring can maximally reduce the immune stimulatory effects of siRNAs [106]. This provides a new insight for the design of anti-miRNAs oligonucleotides to minimize immune stimulatory effects.

Despite great progresses in miRNA drug discovery, to find clinically safe and effective miRNA drugs still remains a huge challenge, and further breakthrough is urgently needed.

## Figures and Tables

**Figure 1 biomolecules-12-00496-f001:**
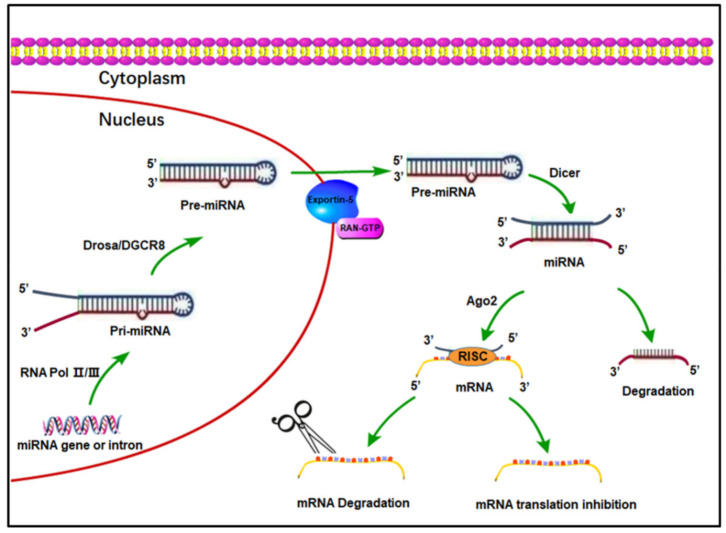
The biogenesis processes of microRNAs (miRNAs). In the nucleus, miRNA genes or introns are transcribed into primary miRNA transcripts (pri-miRNA) by RNA polymerase II/III. Each pri-miRNA contains at least one hairpin structure recognized and processed by the microprocessor complex, Drosha and its partner, DGCR8, which consists of the RNase III type endonuclease. Additionally, the microprocessor complex generates a 70-nucleotide stem loop known as the precursor miRNA (pre-miRNA). Subsequently, pre-miRNAs are exported to the cytoplasm by Exportin 5-RAN-GTP. In the cytoplasm, the pre-miRNA is recognized by Dicer, which consists of RNase III type endonuclease, and then pre-miRNA is cleaved by Dicer to generate a 20-nucleotide mature miRNA duplex. Subsequently, one strand serves as the biologically active mature miRNA, whereas the other one is degraded. At last, the mature miRNA is loaded with Ago2 proteins and incorporated into the RNA-induced silencing complex (RISC). In the RISC, mature miRNA recognizes the target mRNAs through partial sequence complementarity with its target. The RISC can inhibit the expression of the target mRNA through two main mechanisms: mRNA degradation by removal of the polyA tail or translation inhibition.

**Figure 2 biomolecules-12-00496-f002:**
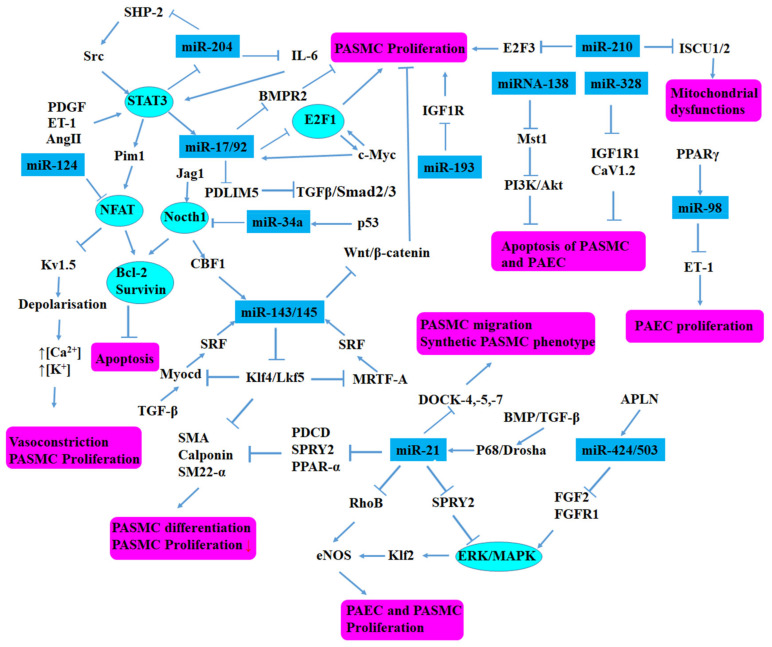
Schematic demonstration of pathways modulated by miRNAs. The changes in miRNAs expression level may lead to the progression of PH. Reproduced and adapted from Grant et al. [46].

**Table 1 biomolecules-12-00496-t001:** Commonly dysregulated miRNAs in PH.

miRNA	Model/Patient Tissue	Change	Effects	PH Targets	References
**miRNAs that Promote Hypoxia or MCT Induced PH**					
miR-143/miR-145	Hypoxia, PH patients	↑	Proliferation ↑	*BMPR2*	[16,17,18,19,20]
miRNA-21	MCT, hypoxia, hypoxia/Sugen5416, lung and serum of PH patients	↑	Proliferation ↑, Migration ↑	*PDCD4, PPARα, RhoB*	[18,21,22,23,24,25]
miRNA-210	PAECs induced by hypoxia	↑	Proliferation ↑	*E2F3, ISCU1/2*	[26,27,28,29,30,31,32]
miRNA-138	PASMCs and PAECs induced by hypoxia	↑	apoptosis ↓	*Mst1*	[33,34]
miRNA-17/92	MCT, hypoxia	↑	NA	*BMPR2, PDLIM5*	[12,35,36]
**miRNAs that prevent PH**					
miRNA-124	Human PASMCs and mouse lungs induced by hypoxia	↓	Proliferation ↓, Migration ↓	*NFATc1, CAMTA1, and PTBP1*	[37,38]
miRNA-328	Rat induced by hypoxia and pulmonary arteries (PA) from PH patients	↓	apoptosis ↑	*IGF1R, CaV1.2*	[39,40]
miRNA-424/503	PAECs from PH patients	↓	Proliferation ↓	*FGF2FGFR1*	[41,42]
miRNA-204	Rat induced by hypoxia or MCT and in human PH patients	↓	Proliferation ↓, apoptosis ↑	*SHP2*	[13,43]
miRNA-98	PAECs from PH patients, PAECs under hypoxia and in lung from mice induced by Sugen5416/hypoxia	↓	Proliferation ↓	*ET-1*	[44]
miRNA-193	MCT, hypoxia	↓	Proliferation ↓	*IGF1R*	[45]

**Abbreviations:** PH, pulmonary hypertension; MCT, monocrotaline; NA, not available; PAECs, pulmonary artery endothelial cells; PASMCs, pulmonary artery smooth muscle cells.

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
