# Peer review of "MicroRNAs in Pulmonary Hypertension, from Pathogenesis to Diagnosis and Treatment"

_biomolecules, 2022, doi:10.3390/biom12040496_

Round 1

Reviewer 1 Report

The manuscript is a comprehensive review of a topical issue where there are still many issues to be resolved. In general, all the interesting aspects related to miRNAs and pulmonary hypertension are treated. It includes a table with the main miRNAs and their actions and a figure also suitable. The bibliography is correct. The main problem is that there are quite a few grammatical errors that need to be corrected.

Perhaps some other miRNAs that could be important in the development of PAH should be included in a review like this.

For example, miRNA-483 (EMBO Mol Med 2020; 12:e11303) that has lower concentration in patients with PAH  and favoring cell proliferation

 Another recent paper has demonstrated a potential role as a therapeutic target for miRNA-146-5p, because this miRNA may increase the proliferation of pulmonary endothelial cells under hypoxia (Dis Markers. 2021; 2021:3668422)

In my opinion, these two references are enough relevant to be included.

Author Response

Dear Sir/Madam,

We have revised the manuscripts according to your suggestion. Thank you for your advice.

We have added miRNA-483 (EMBO Mol Med 2020; 12: e11303) and miRNA-146-5p (Dis Markers. 2021; 2021:3668422) in the “2. miRNAs correlated with PH” section.

Kind regards,

Deming Gou

E-Mail: dmgou@szu.edu.cn

Reviewer 2 Report

This review article addresses topical issues related to the use of some miRNAs as disease markers and potential targets for therapy. The review focuses on those miRNAs that have been associated with the development of pulmonary hypertension. The authors themselves are actively working in this area of research and, of course, are well aware of the subject they are writing about. However, the text of the manuscript needs to be revised before publication.

Abstract: There are many abbreviations in the abstract, which makes it difficult to understand the content of the text.

Line 30:  hypertension(PH) – space needed. In addition, two different abbreviations PN and PAN occur in the text. It's better to use just one of them.

Lines 38-40: The text describing Figure 1 is too general, without the necessary explanation of the details of miRNA processing.

Figure 1: Figure 1 is original, but drawn without the necessary details. For example, there are no indications of 5’ and 3’ ends, no complementarity is observed in the double-stranded part of the pri- and pre-miRNA structures. From figure 1, it can be understood that the double-stranded complex connected to RISK must be attached to the mRNA. It is impossible to understand from the figure how the microRNA chain that is degrading differs from the one that is functionally active.

Lines 61-62: What is written in the second sentence does not follow from the first. Needs to be reformulated.

Lines 63-65: check English, please. There is no predicate in the sentence

Lines 66-67: English problems

 Line 68: English problems and What is MCT?

Lines 69-74: English problems in every sentence

Table 1: Not all microRNAs, which are further described in more detail in the text, are presented in Table 1. What are “PAH targets” in Table 1? If they are genes, then they should be written in Italics

Line 81: It is not clear what the authors mean by the notion that Figure 2 has been adapted? Perhaps the authors wanted to write that figure 2, published in the cited article, was supplemented?

 Line 224: (FRS2[46].  

The manuscript does not even mention long non-coding RNAs, which often act as sponges for microRNAs and, accordingly, take part in the regulation of gene expression levels. It is necessary to add information about the role of lncRNA in the regulation of microRNAs during the development of pulmonary hypertension. Do the authors think that the involvement of lncRNA can influence the emerging problems of diagnostics and the use of miRNAs as therapeutic targets/agents? This issue needs to be discussed.  

There are a lot of abbreviations in the text of the manuscript, and not all abbreviations are deciphered.

 There are many inaccuracies in the expression of the authors' thoughts in the text, and there are many problems with the translation into English

. Just several of them are listed above.  The entire text of the manuscript must be shown to a professional translator for editing. 

Author Response

Dear Sir/Madam,

We have revised the manuscripts according to your suggestion. Thank you for your advice.

we have made all the correction of the manuscript, redrawn the Fig.1, and described the biogenesis processes of microRNAs of Fig.1 in detail, and the effects of long non-coding RNAs on microRNAs expression. We shall describe it point by point as shown below.

(1)   Abstract: the second reviewer pointed out that “There are many abbreviations in the abstract, which makes it difficult to understand the content of the text.”

We have made a modification of the abstract section to ensure it easily to understand the content of the text.

(2)   Line 30: the second reviewer pointed out that “hypertension(PH) – space needed. In addition, two different abbreviations PN and PAN occur in the text. It's better to use just one of them.”

We have added a space in “hypertension (PH)” and unified the two different abbreviations PN and PAN occur in the text except for iPAH (idiopathic pulmonary arterial hypertension).

(3)   Lines 38-40: the second reviewer pointed out that “The text describing Figure 1 is too general, without the necessary explanation of the details of miRNA processing.”

We have described the processes of miRNAs in detail as shown in the recently submitted manuscript.

(4)   Figure 1: the second reviewer pointed out that “Figure 1 is original, but drawn without the necessary details. For example, there are no indications of 5’ and 3’ ends, no complementarity is observed in the double-stranded part of the pri- and pre-miRNA structures. From figure 1, it can be understood that the double-stranded complex connected to RISK must be attached to the mRNA. It is impossible to understand from the figure how the microRNA chain that is degrading differs from the one that is functionally active.”

We have added indications of 5’ and 3’ ends of miRNAs and mRNA, and complementarity in the double-stranded part of the pri- and pre-miRNA structures. Moreover, we make it easily to understand how one strand of the mature miRNA is loaded with Ago2 proteins and incorporated into the RNA-induced silencing complex (RISC). Furthermore, we make it discernable between the strands of the microRNA chain that is degrading and the one that is functionally active.

(5) Lines 61-62: the second reviewer pointed out that “What is written in the second sentence does not follow from the first. Needs to be reformulated.”

We have reformulated the second sentence as “And changes in miRNA expression level can lead to various cardiovascular disorders, including atherosclerosis, peripheral vascular disease, and PH.”

(6) Lines 63-65: the second reviewer pointed out that “check English, please. There is no predicate in the sentence.”

We have checked and added predicate in the sentence.

(7) Lines 66-67: the second reviewer indicated that there were some “English problems”.

 We have added a predicate and solved the problems.

(8) Line 68: the second reviewer indicated that there were “English problems and What is MCT?”

For first, we have modified the “reduce” for “be reduced” to solve the problems. Subsequently, we give clear indication of MCT (monocrotaline).

(9) Lines 69-74: the second reviewer indicated that there were “English problems in every sentence”. We have modified this section: “Furthermore, our group has found that miR-223, miR-328, miR-339, miR-4632, miR-1181 and miR-1281 were down-regulated and prevented the development of PH. In contrast, miR-20a and miR-125a-5p were up-regulated and promoted the development of PH”.

(10) Table 1: the second reviewer indicated that “Not all microRNAs, which are further described in more detail in the text, are presented in Table 1. What are “PAH targets” in Table 1? If they are genes, then they should be written in Italics”.

We have only chosen parts of attractive miRNAs to demonstrate on Table 1. And “PH targets” in Table 1 means PH target gene and they are written in Italics.

(11) Line 81: the second reviewer indicated that “It is not clear what the authors mean by the notion that Figure 2 has been adapted? Perhaps the authors wanted to write that figure 2, published in the cited article, was supplemented?”

The “adapted” means that parts of the contents in the figure was presented in the quoted reference, then we have made some revision and supplement.

(12) Line 224: the second reviewer indicated that it lacked a parenthesis on the right of “(FRS2”. We have added a parenthesis on the right.

(13) The second reviewer indicated that “The manuscript does not even mention long non-coding RNAs, which often act as sponges for microRNAs and, accordingly, take part in the regulation of gene expression levels. It is necessary to add information about the role of lncRNA in the regulation of microRNAs during the development of pulmonary hypertension. Do the authors think that the involvement of lncRNA can influence the emerging problems of diagnostics and the use of miRNAs as therapeutic targets/agents? This issue needs to be discussed.”   

We have added a section “5. Regulation of miRNAs by lncRNAs in PH” to probe into regulation of lncRNAs in PH, and the role of lncRNA in the regulation of microRNAs during the development of PH. Moreover, we have also briefly discussed that the involvement of lncRNA can influence the emerging problems of diagnostics and the use of miRNAs as therapeutic targets/agents.

 (14) The second reviewer indicated that “There are a lot of abbreviations in the text of the manuscript, and not all abbreviations are deciphered.”

We have added lots of abbreviations in the table of Abbreviations in order to cover all nonstandard abbreviations in the manuscript.

(15) The second reviewer indicated that “There are many inaccuracies in the expression of the authors' thoughts in the text, and there are many problems with the translation into English.”

We have invited an experienced interpreter who is native in English and made a modification of the manuscript. Moreover, we have also examined the manuscript in detail. The details of the revision have been presented in the recently submitted manuscript.

Kind regards

Deming Gou

E-Mail: dmgou@szu.edu.cn

Round 2

Reviewer 2 Report

The authors answered all questions and made the necessary corrections to the text of the manuscript, including significantly improving English. However, there are still comments on the quality of the English translation of the newly written text of the manuscript. In addition, there are still a number of comments that need to be corrected before publication (see below).

Lines 69-70: at least one hairpin structures – check English, please

Lines 121-122: can also induced – check English, please

Line 221: 404/503 – did you mean miR-424/503?

Lines 285-313: The text has many problems with translation into English

LnRPT (line 299) and lnRPT (line 301) -  ­It's not clear why the spelling is different

Line 303: competitive RNA (ceRNA)  – did you mean competing endogenous RNA?

 Lines 302-313: This text is not about hypertension. Please refer to literature on hypertension.

 Line 347: PAH

Line 398: PAH

Line 456: list of abbreviations must be done in alphabetical order

Author Response

For the second submission, the reviewer advised us to make some modification on our manuscript. We have made all the correction of the manuscript. We shall describe it point by point as shown below.

  • Lines 69-70: the reviewer pointed out that “at least one hairpin structures – check English, please.” We have checked and corrected it.
  • Lines 121-122: the reviewer pointed out that “can also induced – check English, please.” We have checked and corrected it.
  • Line 221: 404/503: the reviewer pointed out that “did you mean miR-424/503? It's not clear why the spelling is different for miR-424/503.” Sorry, we made a mistake, and we corrected it for “miR-424/503”.
  • Lines 285-313:

The reviewer pointed out that “The text has many problems with translation into English.” We have invited an experienced interpreter who is native in English and made a modification of the manuscript. Moreover, we have also examined the manuscript in detail. The details of the revision have been presented in the recently submitted manuscript.

The reviewer pointed out that “LnRPT (line 299) and lnRPT (line 301). It's not clear why the spelling is different.” Sorry, we made a mistake for “lnRPT”, and we corrected it for “LnRPT”.

The reviewer pointed out that “competitive RNA (ceRNA) -did you mean competing endogenous RNA?”Yes, we mean competing endogenous RNA and we correct it.

The reviewer pointed out that “The reviewer pointed out that “This text is not about hypertension. Please refer to literature on hypertension. We change the references for hypertension.

  • Line 347 and Line 398: the reviewer pointed out that “PAH”. We have changed “PAH” for “PH”.
  • Line 456: the reviewer pointed out that “list of abbreviations must be done in alphabetical order”. We have done it in alphabetical order.
